# Separation, Purification, Structure Analysis, In Vitro Antioxidant Activity and circRNA-miRNA-mRNA Regulatory Network on PRV-Infected RAW264.7 Cells of a Polysaccharide Derived from *Arthrospira platensis*

**DOI:** 10.3390/antiox10111689

**Published:** 2021-10-26

**Authors:** Mi-Xia Cao, Xiao-Dong Xie, Xin-Rui Wang, Wen-Yue Hu, Yi Zhao, Qi Chen, Lu Ji, Ying-Yi Wei, Mei-Ling Yu, Ting-Jun Hu

**Affiliations:** 1College of Animal Science and Technology, Guangxi University, Nanning 530004, China; caomixia66668888@st.gxu.edu.cn (M.-X.C.); 1718304007@st.gxu.edu.cn (X.-D.X.); 2018393054@st.gxu.edu.cn (X.-R.W.); zhaoyi@st.gxu.edu.cn (Y.Z.); chenqi@st.gxu.edu.cn (Q.C.); 2018393025@st.gxu.edu.cn (L.J.); weiyingyi@gxu.edu.cn (Y.-Y.W.); yumeiling@gxu.edu.cn (M.-L.Y.); 2School of Life Sciences and Biotechnology, Shanghai Jiao Tong University, Shanghai 200030, China; summer86artemis@sjtu.edu.cn

**Keywords:** *Arthrospira platensis* polysaccharide, pseudorabies virus, antioxidant, structure, circRNA-miRNA-mRNA

## Abstract

To investigate the structure of *Arthrospira platensis* polysaccharide (PAP) (intracellular polysaccharide) and the antioxidant activity of the first component of PAP (PAP-1) on pseudorabies virus (PRV) -infected RAW264.7 cells. The PAP was separated and purified by the Cellulose DE-52 chromatography column and Sephacryl S-200 high-resolution gel column to obtain PAP-1. The antioxidant activity and regulation of PAP-1 on PRV-infected RAW264.7 cells of circRNA-miRNA-mRNA network were investigated by chemical kit, Q-PCR, and ce-RNA seq. The results indicated that the molecular weight (Mw) of PAP-1, which was mainly composed of glucose and eight other monosaccharides, was 1.48 × 10^6^ Da. The main glycosidic bond structure of PAP-1 was →4)-α-D-Glcp-(1→. PAP-1 may be increased the antioxidant capacity by regulating the circRNA-miRNA-mRNA network in PRV-infected RAW264.7 cells. This study provided a scientific foundation for further exploring the antioxidant activity of PAP-1 based on its structure.

## 1. Introduction

Algal polysaccharides are natural macromolecular compounds extracted from algal and are composed of more than ten monosaccharide molecules connected by glycosidic bonds. The most commonly used method for extracting polysaccharides is the hot water extraction method, but it requires a long extraction time, high temperature, and a large amount of solvent. Therefore, researchers have improved extraction methods based on the hot water extraction method, including enzyme-assisted hot water extraction method, ultrasonic-assisted hot water extraction method, microwave-assisted hot water extraction method, freeze-thaw-assisted hot water extraction method, alkaline extraction, freeze-thawing cold-pressing, and so on [1,2]. However, different extraction methods not only affect the extraction rate but also influence the structure and biological activity of polysaccharides [2,3]. Therefore, it is necessary to screen the extraction methods to determine which is best and to ensure the structural integrity of polysaccharides when further studying their structures. In this study, the best extraction method of PAP was selected for subsequent experiments, which provided an experimental basis for further study on the structure and activity of PAP.

At present, *Arthrospira* is recommended by FAO as “the most ideal food in the 21st century”, while the World Health Organization calls it “the best health care product for mankind in the 21st century” [4]. PAP, as a natural health product, has no toxic and side effects and can be taken for a long time. PAP is a type of cyanobacteria, which are considered to be one of the most potent species for prevention and nutritious supplementary because of their abundant special active ingredients [5]. At present, researchers on PAP have made progress in the separation, purification, and determination of chemical structure and biological activity, but there are still many problems to be solved. Most studies have been based on crude polysaccharides, but crude polysaccharides may contain multiple mixed sugar components and small molecular impurities, so it is difficult to obtain a definitive conclusion from these studies [6,7,8] In addition, there have been some studies on the structure of isolated and purified PAP by FT-IR, NMR, and GC-MS, but research on the structure has not been thorough enough [5,9,10]. The detailed structure of polysaccharides was the foundation for studying their potential mechanisms and structure-activity relationships. In our research, the structure of PAP-1 is studied not only by FT-IR, NMR, and GC-MS, but also by specific optical rotation, periodate oxidation, smith degradation, and methylation analysis.

PRV is a double-stranded DNA herpes virus that can cause highly contagious neurological and respiratory diseases in swine and is often fatal [11]. It has caused huge economic losses to the swine industry in many countries. Most viruses, such as coronavirus disease 2019 (COVID-19), influenza virus, porcine epidemic diarrhea virus, and hepatitis C virus, cause oxidative stress when they infect cells [12,13,14]. High levels of reactive oxygen species (ROS) can cause serious damage to biological structures, such as cell membrane damage, DNA degeneration, and chromosome translocation. The oxidative stress caused by ROS can cause some diseases [15]. In addition to developing drug therapy, it may be a useful strategy to use dietary supplements and nutritional drugs to prevent or treat SARS-CoV-2 infection in the field of alternative and adjuvant therapy [16]. Bioactive substances in many naturally occurring foods have strong antioxidant activity. Some studies have shown that polysaccharides can reduce oxidative stress caused by adverse stimulation [17,18,19]. Therefore, it is still necessary to study antioxidation food and its mechanisms. Herein, PAP (intracellular polysaccharide) [20] was isolated and purified, and the effect of PAP-1 on the activity and antioxidant capacity of PRV-infected RAW264.7 cells was further investigated.

MiRNA (MicroRNA) is an endogenous single-stranded small molecular RNA with a length of about 22 nucleotides, which is widely distributed in eukaryotes, animals, and plants, and is located in the non-coding region of the genome [21]. MiRNA is a very important regulatory factor of gene expression, which mainly regulates the stability and translation efficiency of its target gene (mRNA). There are two main ways of regulation: Firstly, it is directly and completely complementary to mRNA, which leads to the degradation of the target gene (mRNA) [22]. Secondly, it can induce the target gene (mRNA) to degrade or repress its translation by forming incomplete base complementation with the 3’UTR of the target gene [23]. Some studies suggest that the Japanese encephalitis virus, dengue virus, porcine epidemic strains virus (PEDV), and other viruses could induce a significant variation of miRNA in the host, and miRNA not only regulates virus replication and pathogenicity but also adjusts the immune response of the host [24,25,26]. CircRNA, a closed-loop RNA molecule, is the product of gene cutting, but it participates in gene transcription and protein translation and plays an important role in the occurrence and development of tumors, neurodegenerative diseases, cardiovascular diseases, diabetes, nephropathy, arthritis, viruses, and other diseases. Therefore, it is expected to become a potential new target for diagnosis and treatment. Among its complex biological regulation functions, it mainly plays a role by competing for endogenous RNA (ceRNA): circRNA indirectly regulates the expression of the target gene of miRNA by competitively binding this miRNA [27]. Studies have shown that Chinese medicine may play an active role in the treatment of cardiovascular diseases through circRNA/miRNA/mRNA regulatory network [28]. With the continuous development of sequencing technology, RNA-Seq has become an effective tool for excavating and identifying new transcripts. The above studies indicate that PAP has antiviral and antioxidant effects [5,8,9], and PRV infection RAW264.7 cells may cause the differential expression of circRNA, miRNA, and mRNA [29,30,31]. In this study, the non-coding RNA sequencing data of PRV-infected RAW264.7 cells and PAP-1 acting on PRV-infected RAW264.7 cells from the GEO database were analyzed, and the circRNA-miRNA-mRNA network was successfully constructed, which has the characteristics of highlighting specific molecular functions and mechanisms. Then, functional enrichment analysis and annotation were carried out to further explore the potential role of circRNA, miRNA, and mRNA in the regulation of PAP-1 on PRV-infected RAW264.7 cells. The research is of great significance for PAP-1 to regulate the antioxidant capacity and circRNA-miRNA-mRNA network caused by PRV-infected RAW264.7 cells.

## 2. Materials and Methods

### 2.1. Materials and Chemicals

*Arthrospira platensis* (The production base of *Arthrospira platensis* can produce 500 tons of high-quality *Arthrospira* powder per 100,000 square meters of the breeding area.) was purchased from Beihai Shengbada Biotechnology Co., Ltd., (Beihai, China); Papain, Salicylic acid, Vc, DPPH, and Cellulose DEAE-52 were purchased from Beijing Solarbio Science & Technology Co., Ltd., (Beijing, China); Sephacryl S-200 High Resolution and Dextran standards purchased from Sigma-Aldrich (Shanghai) Trading Co., Ltd., (Shanghai, China); BI fetal bovine serum was purchased from Biological Industries Israel Beit Hemek Ltd., (Nazareth, Israel); RPMI 1640 complete culture medium was purchased from ThermoFisher Biochemical Products (Beijing) Co., Ltd., (Beijing, China); CCK-8 kit was purchased from Beyotime (Shanghai, China); the Reactive Oxygen Species Assay Kit, Malondialdehyde (MDA) assay kit, Myeloperoxidase assay kit, Xanthine Oxidase (XOD) assay kit, Superoxide Dismutase (SOD) assay kit, Catalase (CAT) assay kit, and Glutathione Peroxidase (GSH-Px) assay kit were purchased from Nanjing Jiancheng Bioengineering Institute (Nanjing, China). RAW264.7 cells were purchased from the cell bank of Wuhan University. The PRV-GXLB-2013 strain was isolated, identified, and preserved by the Department of Preventive Veterinary Medicine, College of Animal Science and Technology, Guangxi University.

### 2.2. Comparison of Different Extraction Methods

#### 2.2.1. Different Extraction Methods of PAP

##### 2.2.1.1. Hot Water Extraction Method

*Arthrospira platensis* was extracted with distilled water at 80 °C for 4 h. Trichloroacetic acid was dripped into the above solution to remove the protein in the extract. Then, the supernatant was obtained by centrifugation of the above solution and placed at 4 °C for 24 h. After alcohol precipitation and centrifugation, alcohol precipitation was carried out again for 24 h. The solution was centrifuged again, and the precipitate was washed with acetone two times. Finally, the precipitation obtained by centrifuging the above solution was dried to obtain PAP.

##### 2.2.1.2. Enzyme-Assisted Hot Water Extraction Method

Papain (100 mg) was added to *Arthrospira platensis* aqueous solution (This solution contains 20 mg of *Arthrospira*
*platensis* and 200 mL of distilled water.) and heated at 52 °C for 1 h and 100 °C for 0.5 h to inactivate the enzyme. Then, the samples were incubated in a water bath at 80 °C for 4 h, and the subsequent steps were the same as those in Section 2.2.1.1.

##### 2.2.1.3. Ultrasonication-Assisted Hot Water Extraction Method

The *Arthrospira platensis* aqueous solution was treated by ultrasonication for 1 h (50 °C, 320 W). Then, the samples were placed in a water bath at 80 °C for 4 h, and the remaining experimental processes were the same as those described in Section 2.2.1.1.

##### 2.2.1.4. Microwave-Assisted Hot Water Extraction Method

The *Arthrospira platensis* aqueous solution was microwaved at 600 W for 10 min. Then, the water bath was incubated at 80 °C for 4 h, and the remaining experimental processes were the same as those described in Section 2.2.1.1.

##### 2.2.1.5. Freeze-Thaw Assisted Hot Water Extraction Method

The *Arthrospira platensis* aqueous solution was freeze-thawed repeatedly. Then, the samples were placed in a water bath at 80 °C for 4 h, and the remaining experimental processes were the same as those described in Section 2.2.1.1.

#### 2.2.2. Comparison of Extraction Rate and Polysaccharide Content

The weight ratio of PAP to *Arthrospira platensis* was considered the polysaccharide yield of PAP. The polysaccharide content of PAP was determined by the anthrone-sulfuric acid method.

#### 2.2.3. Comparison of Antioxidant Activity

##### 2.2.3.1. Determination of Hydroxyl Radical Scavenging Activity

The scavenging activities of extracts to hydroxyl free radicals were determined by an improved Cumbes-Sironoff method and Fenton method [32]. Polysaccharides and Vc solutions with different mass concentrations (0, 0.125, 0.25, 0.5, 1.0, 2.0 and 4.0 mg/mL) were prepared. Then, 9 mM FeSO_4_ (50 μL), 9 mM salicylic acid-ethanol (50 μL), polysaccharide solution (50 μL), and 8.8 mM H_2_O_2_ (50 μL) were added to 96-well plates. After incubation at 37 °C for 1 h, the absorbance at 510 nm was measured. The scavenging rate was calculated according to the following formula:Hydroxyl radical scavenging activity (%) = [*A*_0_ − (*A_X_* − *A_X_*_0_)]/*A*_0_ × 100%
where *A*_0_ is the absorbance of the blank control, *A_X_* is the absorbance of the sample, and *A_X_*_0_ is the absorbance of the sample without H_2_O_2_ solution.

##### 2.2.3.2. Determination of DPPH Free Radical Scavenging Activity

The DPPH free radical scavenging activities of polysaccharides were determined by a previously reported method [33]. DPPH (0.0082 g) was dissolved in 95% ethanol to make a 100 mL solution. In addition, polysaccharides and Vc solutions were dissolved in DMSO to prepare different concentrations of inhibitor solutions (0, 0.125, 0.25, 0.5, 1.0, 2.0, and 4.0 mg/mL). The DPPH solution (0.5 mL) was added to different concentrations of the inhibitor solution (0.5 mL). The absorbance was measured at 517 nm after reacting in the dark at room temperature for 30 min. The scavenging rate was calculated according to the following formula:DPPH radical scavenging activity (%) = [*A*_0_ − (*A_i_* − *A_i_*_0_)]/*A*_0_ × 100%
where *A*_0_ is the absorbance of the blank control, *A_i_* is the absorbance of the sample, and *A_i_*_0_ is the absorbance of the sample without the DPPH solution.

##### 2.2.3.3. Reducing Power

The method for the determination of reducing power is based on a previously reported method [34]. The polysaccharides and Vc solutions were dissolved in PBS to prepare solutions at different concentrations (0, 0.125, 0.25, 0.5, 1.0, 2.0, and 4.0 mg/mL). Then, the polysaccharide solution (50 μL) and 1% potassium ferricyanide (50 μL) were mixed and incubated at 50 °C for 20 min. After incubation, the mixture was cooled to room temperature, and 10% trichloroacetic acid (40 μL) was added. FeCl_3_ was added to the supernatant, and the absorbance was measured at 700 nm after 5 min. The reducing power was calculated according to the following formula:Reducing power = *A*_1_ − *A*_2_
where *A*_1_ is the absorbance of the sample and *A*_2_ is the absorbance with deionized water instead of FeCl_3_.

### 2.3. Separation and Purification of PAP

PAP extracted by enzyme-assisted hot water extraction method was prepared into 20 mg/mL solution. PAP solution was added to a Cellulose DE-52 chromatography column (2.0 cm × 60 cm) and then subsequently eluted with deionized water, 0.1, 0.2, 0.3, 0.4, and 0.5 M NaCl solution. The eluent was collected in test tubes by fraction collector, each tube was 5 mL. The content of polysaccharides in each tube was detected by the anthrone-sulfuric acid method, and then the elution curve was drawn with the number of tubes as abscissa and the content of polysaccharide as ordinate. The main absorption peaks were mixed according to the polysaccharide content of the eluent. NaCl was removed by dialysis in deionized water, and the purified polysaccharides were obtained by vacuum freeze-drying. Then, each component separated and purified by Cellulose DE-52 chromatography column was further purified by a Sephacryl S-200 high-resolution gel column (1.7 cm × 100 cm). The polysaccharide content of the eluent was also detected, and the absorption peaks were combined and finally freeze-dried.

### 2.4. Comparison of Antioxidant Activity of PAP

The methods of scavenging hydroxyl radicals, DPPH radicals, and reducing power were the same as those described in Section 2.2.3.

### 2.5. Structure Analysis of the First Component of PAP (PAP-1)

#### 2.5.1. Determination of Molecular Weight

Dextran standards (5 mg/mL) with different molecular weights (1152, 5000, 11,600, 23,800, 48,600, 80,900, 148,000, 273,000, 409,800, and 667,800 Da) and PAP-1 were centrifuged and filtered. The purity of PAP-1 was identified, and its molecular weight was determined by HPGPC. The conditions were as follows: Chromatographic column: BRT105-104-102 gel column (8 × 300 mm) (BoRui Saccharide, BRT105-104-102, Yangzhou, China); mobile phase: 0.05 M NaCl; flow rate: 0.6 mL/min; column temperature: 40 °C; sample size: 20 μL; detector: differential detector RI-502 (Shimadzu (China) Co., Ltd., RI-502, Guangzhou, China).

#### 2.5.2. Specific Rotation

The optical intensity of PAP-1 (0.4 mg/mL) was measured by a digital automatic polarimeter (Shanghai Precision Scientific Instruments Co., Ltd., WZZ-2S, Shanghai, China) at 20 °C.

#### 2.5.3. FT-IR

PAP-1 was mixed with KBr and pressed into thin slices, which were scanned by FT-IR (Thermo Scientific, Nicolet iS50, New York, NY, USA) in the range of 4000~400 cm^−1^.

#### 2.5.4. Monosaccharide Composition Analysis

PAP-1 (5 mg) was dissolved in 2 M trifluoroacetic acid (2 mL) and then hydrolyzed at 121 °C for 2 h, but the standard solutions have not been hydrolyzed by trifluoroacetic acid. Nitrogen was introduced into the acidolysis solution and blown dry. The powder was cleaned with methanol repeatedly and blown dry three times to remove the trifluoroacetic acid. The obtained dry powder was dissolved in distilled water and transferred to a chromatographic bottle for determination. Galacturonic acid, glucuronic acid, 6-Deoxy-L-mannosehydrat, arabinose, fucose, mannose, galactose, glucose, fructose, xylose, and ribose were prepared in 1, 10, 20, 30, 40, and 50 μg/mL solutions, respectively. The standard can be directly analyzed on the machine without derivation. The adopted chromatographic system was a Thermo ICS-5000+ ion chromatographic system (Thermo Fisher Scientific, ICS-5000+, New York, NY, USA). The chromatographic conditions were as follows: Dionex™ CarboPac™ PA20 liquid chromatography column; sample size: 20 μL; mobile phase A: H_2_O; mobile phase B: 100 mM NaOH; column temperature: 30 °C. The monosaccharide components were detected by the electrochemical detector (HPLC-DAD) (Shimadzu (China) Co., Ltd., Shimadzu LC-10A, Guangzhou, China).

#### 2.5.5. Periodate Oxidation and Smith Degradation

PAP-1 (50 mg) was oxidized with 15 mM NaIO_4_ (25 mL) and then reacted at 4 °C in the dark. The absorbance was measured at 223 nm every 4 h until it reached a stable value. The consumption of periodic acid was calculated according to the standard curve of NaIO_4_. The yield of formic acid was determined by titration with 0.053 M NaOH. Then, ethylene glycol was added to terminate the periodic acid reaction. The oxidized products of periodate were dialyzed in water for 48 h and reduced by adding NaBH_4_ (50 mg) for 12 h. Then, acetic acid was added to adjust the pH, and the solution was dialyzed in deionized water for 48 h. The residue was completely hydrolyzed with 2 M trifluoroacetic acid. Methanol was repeatedly added and dried to remove trifluoroacetic acid. Finally, the product was acetylated and analyzed by GC (Agilent Technologies lnc., Agilent 7820A; Santa Clara, CA, USA) [35].

#### 2.5.6. Methylation Analysis

PAP-1 (1 mg) was methylated so that all the free hydroxyl groups were methylated by the Hakomori method. Infrared spectroscopy is used to detect the absorption at 3500 cm^−1^ to determine whether the methylation reaction is complete. Then, 2 M trifluoroacetic acid (100 μL) was added and hydrolyzed at 121 °C for 90 min to obtain a partially methylated monosaccharide. These monosaccharides were reduced and acetylated, and the methylene chloride phase in the lower layer contained partially methylated alditol acetate. The above derivatives were analyzed by GC-MS, and the connection mode of the glycosidic bond was obtained by analyzing the peak sequence in GC and the main ion fragments of MS. Mass spectrometry conditions were as follows: PAP-1 was detected by electron bombardment ion source (EI) in full SCAN mode, and the mass scanning range (m/z) was 30-600 (Agilent Technologies lnc., Agilent 5977B; Santa Clara, CA, USA). Gas chromatography conditions: 140 °C for 2.0 min, 3 °C/min to 230 °C for 3 min, and the injection volume was 1 μL (Agilent Technologies lnc., Agilent 7820A; Santa Clara, CA, USA).

#### 2.5.7. NMR Analysis

PAP-1 (50 mg) was dissolved in D_2_O (0.5 mL) and freeze-dried into powder (this process was repeated to fully exchange active hydrogen). The ^1^H-NMR, ^13^C-NMR, one-dimensional spectrum, and two-dimensional spectrum of DEPT135 in which PAP-1 was dissolved in D_2_O were measured by a 600 MHz NMR spectrometer (Bruker Scientific Technology (Shanghai) Co., Ltd., Bruker AVANCE III HD600, Shanghai, China) at 25 °C.

### 2.6. The Effect of PAP-1 on Antioxidation of PRV-Infected RAW264.7 Cells

#### 2.6.1. The Effect of PAP-1 on the Activity of RAW264.7 Cells

RAW264.7 cells at a concentration of 1 × 10^6^ cell/mL were added to 96-well culture plates. They were divided into a blank group, cell group, and PAP-1 groups (25, 50, 100, 200, 400, 800, and 1600 μg/mL). Culture medium (100 μL) was added to the cell group, PAP-1 solution at different concentrations was added to the PAP-1 groups, and the cells were cultured for 8 h, 12 h, 24 h, or 48 h. Cell counting kit-8 (CCK-8) assay was used to detect cell activity [36]. Add the CCK-8 solution (10 μL) to each well of 96-well plates, then put them in an incubator to incubate for 1–4 h, and finally use a microplate reader (Tecan (Shanghai) Trading Co., Ltd., Infinite M200 Pro, Zürich, Switzerland) to detect the absorbance at 450 nm.

#### 2.6.2. The Effect of PAP-1 on the Activity of PRV-Infected RAW264.7 Cells

The following groups were used: blank group, cell group, PRV group, and PAP-1 groups (25, 50, 100, 200, and 400 μg/mL). Cell culture medium (100 μL) was added to the cell group, and PRV solution (100 μL) was added to the other groups and incubated for 2 h in 96-well culture plates. The supernatant was discarded, and the cells were washed with PBS three times. Culture medium was added to the cell group and PRV group, while PAP-1 solution (100 μL) was added to the PAP-1 groups. They were cultured for 8 h, 12 h, 24 h, or 48 h, and the cell activity was detected by the CCK-8 method.

#### 2.6.3. The Effect of PAP-1 on the Antioxidative Capacity in PRV-Infected RAW264.7 Cells

This experiment included the cell group, PRV group, and PAP-1 groups (50, 100, and 200 μg/mL) (96-well culture plates). The steps of incubating PRV and adding PAP-1 solution were the same as those in Section 2.6.2. The supernatant was discarded, and the cells were washed with PBS three times. DCFH-DA fluorescent probe (100 μL) was added to each well and incubated for 30 min. Finally, PBS (100 μL) was added to each well to remove the cells, and then the absorbance was measured at an excitation wavelength of 488 nm and an emission wavelength of 525 nm according to the Reactive Oxygen Species Assay Kit.

This experiment also included the cell group, PRV group, and PAP-1 groups (50, 100, and 200 μg/mL) (6-well culture plates). The steps of incubating PRV and adding PAP-1 solution were the same as those in Section 2.6.2. The supernatant was used to measure MDA level, and the cells were washed with PBS three times. Add PBS (500 μL) to each well, scrape off the cells, crush the cells with a cell ultrasonic breaker (Ningbo Scientz Biotechnology Co., Ltd., SCIENTZ-IID, Ningbo, China), centrifuge at 3000 rpm for 10 min, and then take the supernatant to measure the activities of GSH-Px, SOD, MPO, XOD, and CAT in the cells. The experiment was carried out in strict accordance with the instructions of the kits.

#### 2.6.4. CeRNA-Seq

CircRNA-miRNA-mRNA regulatory network was constructed by ceRNA-seq to further explore the antioxidant regulatory effect of PAP-1 on PRV-infected RAW264.7 cells. The experiment was divided into three groups, which were PAP-1 group (200 μg/mL), PRV group, and cell group, with three replicates in each group (When performing ceRNA-Seq, each group was labeled for convenience. “C” is the abbreviation of cell, “V” is the abbreviation of virus, and “M” is the abbreviation of medicine. Therefore, the PAP-1 group, PRV group, and cell group are marked as CVM group, CV group, and C group, respectively). RAW264.7 cells at a concentration of 1 × 10^6^ cells/mL were added into a 6-well plate. After incubation for 8 h at 37 °C with 5% CO_2_, PRV solution was added and incubated for 2 h. The virus solution was discarded and washed with PBS three times. Finally, PAP-1 was added and incubated for 12 h. Cell samples were collected, and the ceRNA-seq was completed by Guangzhou Gidio Biotechnology Co., Ltd. The constructed sequencing library was sequenced with Illumina HiSeqTM 4000 to find the differentially expressed circRNA, miRNA, and mRNA among the three groups of PAP-1 group, PRV group, and cell group. According to the sequence of circRNA, the targetScan 7.2 is used to predict the interaction with miRNA, and then combined with the regulatory relationship between miRNA and genes in the miRTarBase and miRBase databases, Cytoscape 3.7.1 software is used to construct a circRNA-miRNA-mRNA regulatory network. Finally, Gene Ontology and KO enrichment analysis were performed on the mRNA differentially expressed in the C group and the CV group, and the CV group and the CVM group.

#### 2.6.5. Quantitative Real-Time PCR

RAW264.7 cells treatment was the same as the second experiment in Section 2.6.3. The total RNA of RAW264.7 cells was extracted by Trizol reagent according to the manufacturer’s instructions. cDNA was synthesized with the 5×All-In-One RT MasterMix with AccuRT (Lot. 0229854828001, abm). Quantitative real-time PCR EvaGreen 2 × qPCR MasterMix-No Dye (Lot. 10518551127002, abm). The mRNA expression levels of MPO, XOD, SOD, and GSH-Px were detected by Q-PCR.

### 2.7. Statistical Analysis

The data were analyzed by SPSS 22.0, and the results include the mean ± SD. One-way analysis of variance (ANOVA) was used for comparisons among groups, and the LSD-t-test was used for pairwise comparisons among groups.

## 3. Results and Discussion

### 3.1. Comparison of Different Extraction Methods

In this study, the extraction rate and contents of polysaccharides processed by the three extraction methods except for the microwave-assisted hot water extraction method were significantly higher than those processed by the hot water extraction method (Figure 1A,B) (*p* < 0.05 or *p* < 0.01). Scavenging hydroxyl radicals, DPPH radicals, and reducing power are important indexes to evaluate the antioxidant activity of polysaccharides [37]. The scavenging capacity of PAP increased rapidly and then slowly with increasing concentration, which was consistent with a previous research method [38]. The order of scavenging of hydroxyl radicals by different extraction methods, DPPH radicals, and reducing power from strong to weak was enzyme-assisted hot water extraction method > freeze-thaw assisted hot water extraction method > microwave-assisted hot water extraction method > ultrasonic-assisted hot water extraction method > hot water extraction method (Figure 1C–E). Compared with other methods, the enzyme-assisted hot water extraction method had the advantages of mild reaction conditions, high extraction rate, less damage to polysaccharide structure, avoid changing the biological activity of polysaccharides, low cost, energy savings, and environmental protection [39]. Therefore, the PAP obtained by the enzyme-assisted hot water extraction method was selected for subsequent experiments.

### 3.2. Isolation and Purification

The PAP obtained by the enzyme-assisted hot water extraction method was separated into five peaks by a Cellulose DE-52 chromatography column; that is, it mainly contained five polysaccharide components (PAPs) (Figure 2A). After further purification by a Sephacryl S-200 High-Resolution chromatography column, each component showed a single symmetrical elution peak, indicating that the molecular weights of PAPs were uniform (Figure 2B–F). The polysaccharide contents of PAPs were significantly higher than those of PAP (*p* < 0.01) (Figure 2G), and the results were similar to that of Ren et al. [40]. The yield of PAP-1 was the highest of the PAPs, so the structure of PAP-1 was analyzed in detail in this study.

### 3.3. Determination of the Antioxidant Activity of PAPs

From Figure 3A–C, the antioxidant activities of PAPs showed a dose-dependent relationship with the concentration. The order of scavenging hydroxyl radicals, DPPH radicals, and reducing power from strong to weak was PAP-3 > PAP-1 > PAP-4 > PAP-5 > PAP-2. Overall, the antioxidant activities of PAPs were different; this trend is consistent with the results of Chen et al. [38].

### 3.4. Structure Analysis of PAP-1

#### 3.4.1. Molecular Weight

From Figure 4A, the equation of the lgMw-RT calibration curve was y = −0.206x + 12.862 (R^2^ = 0.9919). The Mw of PAP-1 was calculated by the formula to be 1.48 × 10^6^ Da.

#### 3.4.2. Specific Optical Rotation

The specific rotation [α]^20^_D_ of PAP-1 at 20 °C was +196.5° (c = 0.4 mg/mL, H_2_O). The higher positive value of optical rotation indicated that PAP-1 mainly exhibits α-glycosidic bonding [41].

#### 3.4.3. FT-IR

In Figure 4B, there was a strong absorption peak at 3350.82 cm^−1^, which was attributed to the stretching vibration of hydroxyl groups. The absorption peak produced by the stretching vibration of C-H appeared at 2929.83 cm^−1^. These two characteristic absorption peaks indicated the presence of polysaccharides in the sample [42]. The peak at 1658.35 cm^−1^ was the hydration vibration peak C-O of polysaccharides [43]. The peak at 1416.54 cm^−1^ was due to the bending vibration of C-H. Three peaks at 1155.99, 1080.76, and 1022.04 cm^−1^ indicated the presence of a pyranose ring and was caused by the C-O-C vibration. The peak of carbohydrate molecules, that is, the asymmetric ring stretching vibration of pyranose, appeared at 932.41 cm^−1^. The absorption peak at 851.95 cm^−1^ indicated that there were α-glycosidic bonds and C-H α-anomers. The peak at 762.41 cm^−1^ was due to the inclusion of C-O-C and was caused by the stretching vibration of the symmetric ring of D-glucopyranose [44]. In conclusion, PAP-1 is an α-configured neutral polysaccharide with a pyranose ring.

#### 3.4.4. Analysis of Monosaccharide Composition

The molar ratio of the monosaccharide composition of PAP-1 was: fucose:6-Deoxy-L-mannosehydrat:arabinose:galactose:glucose:xylose:mannose:galacturonic acid:glucuronic acid = 0.15:0.15:0.08:0.02:54.99:0.06:0.04:0.01:0.02 (Figure 4C,D). Chaiklahan et al. found that the content of rhamnose in PAP was the highest, but our results showed that the content of glucose was the highest in PAP-1, which may be due to the monosaccharide compositions of PAP being related to its place of origin [45].

#### 3.4.5. Periodate Oxidation and Smith Degradation

After PAP-1 was oxidized by periodic acid, the consumption of periodic acid and the formation of formic acid were 1.02 mM and 0.08 mM, respectively. The oxidized samples were subjected to smith degradation, and then the products were reduced. The products were mainly erythritol and a small amount of glycerol, which indicated that there were glucose residues linked by (1→4) glycosidic bonds in PAP-1.

#### 3.4.6. Methylation Analysis

Three kinds of connection modes (t-Glc(p); 4-Glc(p); 4,6-Glc(p)) and three derivatives (1,5-di-O-acetyl-2,3,4,6-tetra-O-methyl glucitol; 1,4,5-tri-O-acetyl-2,3,6-tri-O-methyl glucitol; 1,4,5,6-tetra-O-acetyl-2,3-di-O-methyl glucitol) were obtained by methylation analysis of PAP-1 (Figure 4E–H).

#### 3.4.7. NMR

The ^1^H NMR spectrum signals of PAP-1 were mainly concentrated between 3.0 and 5.5 ppm. The proton signals of the sugar ring appeared between δ3.2-4.0 ppm, and the signal peaks of the main terminal matrix were present at δ5.31, 5.26, 5.15, 4.89, and 4.57 ppm and were concentrated in the range of 4.3~5.5 ppm (Figure 5A).

From Figure 5B, the signals in the ^13^C NMR (201 MHz, D_2_O) spectrum of PAP-1 were mainly concentrated between 60–120 ppm. By observing the carbon spectrum, it can be seen that the main anomeric carbon signal peaks were mainly located between δ93 and 105 ppm; they appeared at δ101.33, 101.05, 99.88, 97.01, and 93.33, separately. The main signal peaks between 60–80 ppm appeared at δ79.52, 78.39, 78.31, 77.5, 77.4, 77.1, 75.56, 74.99, 74.56, 74.03, 74.02, 72.91, 72.86, 72.8, 72.53, 72.11, 71.94, 71.68, 70.69, 70.61, 68.46, 62.3, 61.89, and 61.67 ppm. These results, combined with monosaccharide composition analysis, showed that PAP-1 mainly contained glucose. According to the analysis of the DEPT135 NMR spectrum, the peaks at 61.67, 62.3, 61.89, and 68.46 ppm were inverted peaks, indicating that they were C6 chemical shifts.

The anomeric carbon signal was at δ101.05, and the corresponding anomeric hydrogen signal was δ5.32 in the HQSC spectrum (Figure 5C). Through the HH-COSY spectrum (Figure 5D), the signals of H1-2 were at 5.32/3.55, H2-3 was 3.55/3.90 and H3-4 was at 3.90/3.58. We can infer that H1, H2, H3, and H4 corresponded to the peaks at δ5.31, 3.55, 3.90, and 3.58, respectively. The TOCSY spectrum showed that the peak at δ5.32 correlated to the peaks at 3.58, 3.78, and 3.90, and H5 was 3.78 ppm (Figure 5E). The corresponding C5 appeared at 72.53, the chemical shift of C6 was δ61.95, and the corresponding H6a was at δ3.79. Therefore, the signal should be attributed to the glycosidic bond →4)-α-Glcp-(1→ [46].

In the HMBC spectrum (Figure 5F), according to the one-dimensional and two-dimensional NMR spectrum, we attributed the glycosidic bond signal of PAP-1. The anomeric hydrogen of glycosidic bond →4)-α-D-Glcp-(1→) had a correlation signal peak with its own C4, and the anomeric carbon had a single correlation peak with its own H4, which indicated that there was a linkage mode of → 4)-α-D-glcp-(1 → 4)-α-D-glcp-(1 →).

In summary, we can infer that the main glycosidic bond structure of the polysaccharide was a →4)-α-D-Glcp-(1→ glycosidic bond. The bonding structures and structural formula of PAP-1 are shown in Figure 5G,H, respectively. Chen et al. showed that the monosaccharides of PAP were mainly composed of rhamnose, fucose, arabinose, xylose, mannose, and glucose with the molar ratio of 3.42:0.76:0.34:0.53:0.43:0.59 [5]. The results of Pr et al. showed that PAP contained a relatively large proportion of galactose [9]. Ma et al. found that PAP was mainly composed of rhamnose, glucose, and galactose [6]. Similarly, Chaiklahan et al. found that the content of rhamnose in PAP was the highest [45]. Li et al. showed that PAP-1 was mainly composed of glucose, which was similar to our results. The difference was that its structure has a C-6 branched by an α-D-Glcp, and its molecular weight was 93.856 KDa. In addition, they obtained *spirulina* from Fujian Shenliu Health Food Co. (Fujian, China) by alkali liquor extraction, which was different from our purchasing place and extraction method [47]. Perhaps the structures of PAP obtained by different extraction methods in different regions may be similar or different, which were helpful to lay a foundation for further study of its mechanism of action.

### 3.5. The Effect of PAP-1 on the Activity of RAW264.7 Cells

PAP-1 could significantly increase cell activity in the range of 25 μg/mL–400 μg/mL within 8–48 h, while PAP-1 at concentrations of 800 μg/mL and 1600 μg/mL significantly decreased the activity of cells at 48 h (*p* < 0.01) (Figure 6A). Perhaps the concentrations of PAP-1 at 800 μg/mL and 1600 μg/mL were a little high, which caused toxic effects on RAW264.7 cells. Most studies have shown that the polysaccharide components isolated and purified from plant polysaccharides can promote cell proliferation in a suitable concentration range (Excessively low concentration of plant polysaccharide may not significantly improve cell activity, while excessive concentration may cause toxicity to cells and inhibit cell proliferation) [10,48,49]. This study was consistent with the above research results. Therefore, PAP-1 in the concentration range of 25–400 μg/mL was used for subsequent experiments.

### 3.6. The Effect of PAP-1 on the Activity of PRV-Infected RAW264.7 Cells

Macrophages are ancient and conservative immune cells that play a key role in host defense and innate immune responses. They can not only initiate the innate immune response but also help to fight infections and inflammation [50]. Adverse stimulation can decrease the activity of macrophages, while plant polysaccharides can increase cell activity caused by harmful stimulation [51]. In this study, the cell activity of the PRV group was highly significantly lower than that of the cell group at 24 h and 48 h (*p* < 0.01). Compared with the PRV group, PAP-1 in the range of 50–400 μg/mL incubated for 24 h and 48 h significantly increased cell activity (*p* < 0.05 or *p* < 0.01). Among them, PAP-1 groups (50–200 μg/mL) have a better effect, so they were selected for subsequent experiments (Figure 6B).

### 3.7. The Effect of PAP-1 on Antioxidant Capacity in PRV-Infected RAW264.7 Cells

Oxidative stress originates from the imbalance between the oxidant produced by ROS and the endogenous antioxidant. Endogenous and exogenous oxidants are related to disease development [52]. The main component involved in oxidative stress is ROS, which is characterized by oxygen-containing reactive chemicals, such as superoxide anion (O_2_^•−^), hydrogen peroxide (H_2_O_2_), hydroxyl radical (^•^OH) and so on. Excessive ROS levels caused by adverse stimulation will cause lipid peroxidation and then produce MDA [53]. MPO can mediate oxidative stress by promoting the production of ROS and reactive nitrogen (RNS) [54]. Xanthine oxidase (XOD) is a homodimer with a molecular weight of 290 kDa, which widely exists in various tissues from bacteria to humans and mammals. XOD uses dioxygen as its substrate to produce O_2_^•−^ and H_2_O_2_ [55]. SOD is responsible for catalyzing the conversion of O_2_^•−^ into H_2_O_2_, and then GSH-Px and CAT convert H_2_O_2_ into H_2_O [56]. At present, many studies have shown that natural plans can improve the antioxidant activity of hosts and are potent antioxidants, which provide a basis for the development and utilization of antioxidant plants. Natural compounds can alleviate cerebral ischemia injury by targeting the activity of MPO to alleviate oxidative stress [54]. *Passiflflora edulis* rinds can eliminate the ROS generated by stimulated polymorphonuclear neutrophils production and inhibit MPO activity to decrease oxidative stress [57]. Taking the XOD inhibitor from plants may be a promising method to prevent microbial infection, inflammation, hypertension, and ischemia/reperfusion injury caused by excessive production of O_2_^•−^ [55]. *Okra* Polysaccharide can reduce ROS and MDA, and increase SOD, GSH-Px, and CAT in the liver of mouse type 2 diabetes mellitus model, that is, it can reduce blood sugar by relieving oxidative stress [58]. The treatment of *momordica charantia* polysaccharides decreased the level of MDA and increased the activities of SOD and CAT in the hippocampus of Kainic acid-induced epileptic rats, and reduced the neuronal damage in the brain induced by kainic acid, which played a neuroprotective role [59]. From Figure 6C, the ROS level in the PRV group was significantly higher than that in the cell group and the levels of ROS in the PAP-1 groups were significantly lower than those in the PRV group at 12–48 h (*p* < 0.05 or *p* < 0.01). As shown in Figure 7A–F, MDA level, and MPO and XOD activities were significantly increased after PRV-infected RAW264.7 cells, while SOD, CAT, and GSH-Px activities were extremely significantly decreased (*p* < 0.01). PAP-1 (50 μg/mL) could significantly reduce MDA level, MPO activity, and increase SOD and CAT activities (*p* < 0.05 or *p* < 0.01). 100 μg/mL and 200 μg/mL of PAP-1 could significantly reduce the level of MDA, the activities of MPO and XOD, and increase the activities of SOD, CAT, and GSH-Px (*p* < 0.01). The expression levels of MPO and XOD mRNA increased significantly (*p* < 0.01), while the mRNA expression levels of SOD and GSH-Px decreased significantly after PRV infected RAW264.7 cells (*p* < 0.05 or *p* < 0.01). Different concentrations of PAP-1 can significantly reduce the expression levels of MPO and XOD mRNA, and significantly increase the mRNA expression levels of SOD and GSH-Px (*p* < 0.01) (Figure 7G). The results of this study are also consistent with those of the above studies. The results indicated that PAP-1 with different concentrations could protect RAW264.7 cells by improving the antioxidant capacity of PRV-infected RAW264.7 cells. However, the molecular mechanism by which PAP-1 acts on PRV-infected RAW264.7 cells needs further study.

### 3.8. The Effect of PAP-1 on the Network of CircRNA-miRNA-mRNA in PRV-Infected RAW264.7 Cells

More and more evidence show that circRNA is closely related to virus infection, and it may play an important role in the pathogenesis and diagnosis of diseases through miRNA. CircRNA-miRNA-mRNA, a regulatory network, has also become an important direction of Chinese herbal medicine regulation. There are 151 circRNAs, 56 miRNAs, and 341 mRNAs differentially expressed in Orf virus-infected goatskin fibroblast cells [60]. The brains of mice infected with Rabies virus were sequenced, and the constructed circRNA-miRNA-mRNA network was composed of 25 differentially expressed circRNAs, 29 miRNAs, and 264 mRNAs [61]. In this study, there were 17 differentially expressed circRNAs, 117 miRNAs, and 2113 mRNAs between C group and CV group (Figure 8A–C) (*p* < 0.05, FC > 1.5). There are 13 differentially expressed circRNAs, 90 miRNAs, and 175 mRNAs between the CV group and the CVM group (Figure 8D–F) (*p* < 0.05, FC > 1.5). Heat map generated by hierarchical clustering analysis of differentially expressed circRNA (Figure 8G,J), miRNA (Figure 8H,K), and mRNA (Figure 8I,L) showed that all samples were clustered into cell groups (C1, C2, C3), PRV group (CV1, CV2, CV3), and PAP-1 group (CVM1, CVM2, CVM3). The results indicated that PRV-infected RAW264.7 cells and PAP-1 acting on PRV-infected RAW264.7 cells caused the differential expression of circRNA, miRNA, and mRNA. To further analyze the regulatory roles among circRNA, miRNA, and mRNA, Cytoscape 3.7.1 software was used to construct a circRNA-miRNA-mRNA network diagram (Figure 9). Generally, the expression correlation between ceRNAs is negatively correlated, that is, circRNA can down-regulate the expression of target miRNAs, and miRNAs can down-regulate mRNA expression. Therefore, the regulatory effect of PAP-1 on PRV-infected RAW264.7 cells was screened based on the expression correlation of circRNA, miRNA and mRNA. Among them, 10 circRNAs, 30 miRNAs, and 71 mRNAs were involved in network regulation. Compared with the CV group, the mRNA expression levels of Igf1, Gclm, Sqstm1, and Slc7a11 genes related to antioxidant activities in the CVM group were significantly increased. The results showed that PRV-infected RAV264.7 cells caused a change in the regulation of the circRNA-miRNA-mRNA network, and PAP-1 acting on PRV-infected RAW264.7 cells may improve antioxidant activity by regulating the circRNA-miRNA-mRNA network.

To reveal the potential functions of differentially expressed circRNA, we used the Gene Ontology website and Innovative Path Analysis software to enrich the functions and signal pathways of circRNA and miRNA-targeted mRNA. Gene Ontology enrichment analysis includes molecular function, cellular component, biological process. The differentially expressed mRNAs screened from C Group and CV Group (Figure 10A–C) or CV Group and CVM Group (Figure 10D–F) were analyzed for Gene Ontology function enrichment by David database. In the molecular function of CV Group and CVM Group, genes are mainly enriched in peroxidase activity, antioxidant activity, and oxidoreductase activity acting on peroxide as acceptor, etc. Enrichment and screening of KEGG pathway in C group and CV group resulted in 315 signaling pathways, involving osteogenic class differentiation, c-type lectin receptor signaling pathway, MAPK signaling pathway, Viral carcinogenesis, TNF signaling pathway, herpes simplex infection, nod-like receptor signaling pathway, microRNAs in cancer, Malaria et al. (Figure 11A–D). 171 signal pathways were obtained by enrichment and screening of the KEGG pathway in the CV Group and the CVM Group, which mainly involved systemic lupus erythematosus, alcoholism, necroposis, viral carcinogenesis, fluid shear stress and atheroslerosis, transcriptional misregulation in cancers, and others signaling pathways (Figure 11E–H). Chicken infected with ALV-J also caused the differential expression of circRNA, in which circRNA_3079 and predicted target genes were mostly concentrated in immune or tumor-related signaling pathways, such as p53 signaling pathway, JAK-stat signaling pathway, nod-like receptor signaling pathway, and other signaling pathways [62]. In addition, total saponins from the leaves of *Panax*
*notoginseng* saponins can regulate chronic unpredictable mild stress. In the model group, the ventral median prefrontal cortex and hippocampus of C57BL male mice expressed a large amount of circRNA, which may be an important mediator of the antidepressant effect of *Panax*
*notoginseng* saponins [63]. The abnormal expression of circRNA may lead to defective or abnormal cell functions, resulting in a variety of human diseases. At the same time, circRNA can be used as a good biomarker because of its structural stability and high tissue specificity [64,65], so it may become a more accurate and effective target for the diagnosis and treatment of diseases in the future.

In conclusion, ceRNA-seq showed that PAP-1 increased the expression of antioxidation-related genes in PRV-infected RAW264.7 cells by regulating the circRNA-miRNA-mRNA network, and caused changes in antioxidation-related signaling pathways. This study can provide a research basis for circRNA-miRNA-mRNA as a drug target to improve the antioxidant capacity of virus-infected cells.

## 4. Conclusions

PAP obtained from the enzyme-assisted hot water extraction method was isolated and purified, and the structure of PAP-1 was deeply studied. The main glycosidic bond structure of PAP-1 was →4)-α-D-Glcp-(1→. PAP-1 could increase the antioxidant capacity of PRV-infected RAW264.7 cells by regulating oxidation and antioxidant factors (MDA, MPO, XOD, SOD, GSH-Px, and CAT). In addition, the circRNA-miRNA-mRNA regulatory network was constructed, and the biological functions and regulatory pathways of differentially expressed mRNA in PRV-infected RAW264.7 cells incubated with PAP-1 were analyzed. We found that PAP-1 may be increased the antioxidant activity of PRV-infected RAW264.7 cells by regulating circRNA-miRNA-mRNA networks and antioxidation-related signaling pathways. This study can lay a certain experimental foundation and open up a new idea for elucidating the activity of PAP-1 from the structure.

## Figures and Tables

**Figure 1 antioxidants-10-01689-f001:**
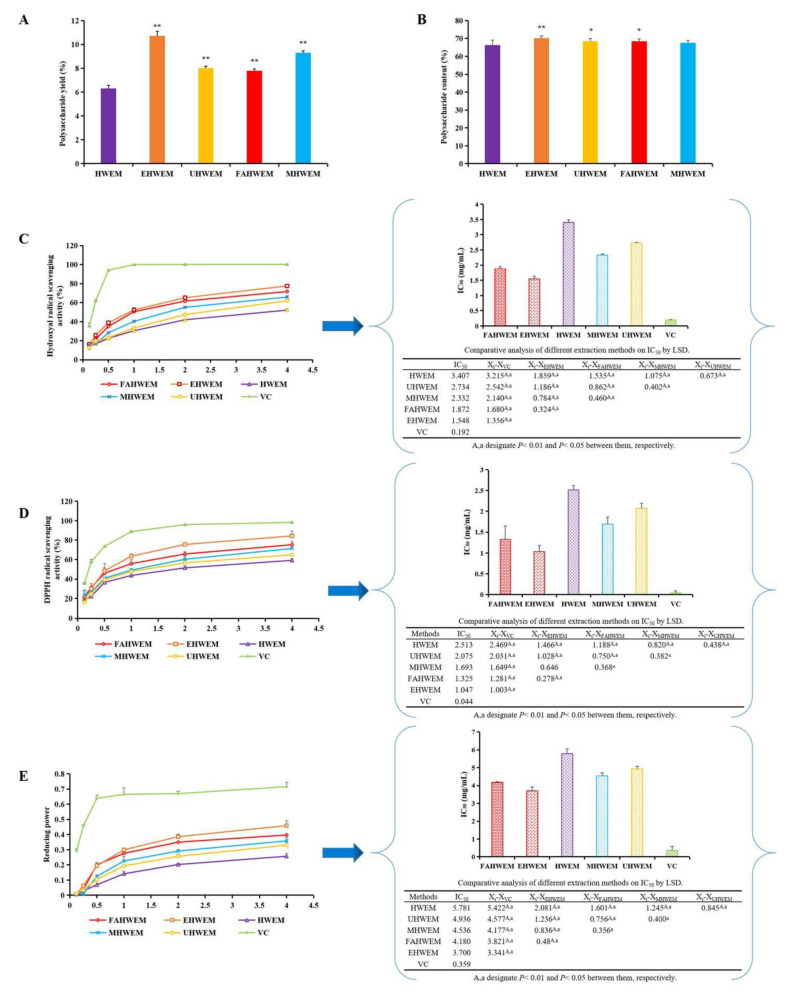
Comparison of polysaccharide yield (**A**) (means ± SD, *n* = 9), polysaccharide content (**B**) (means ± SD, *n* = 9), and antioxidant activity (**C**–**E**) (means ± SD, *n* = 3) of PAP were obtained by different extraction methods. HWEM, hot water extraction method; EHWEM, enzyme-assisted hot water extraction method; UHWEM, ultrasonic-assisted hot water extraction; MHWEM, microwave-assisted hot water extraction; FAHWEM, freeze-thaw assisted hot water extraction. Bars with * and ** indicate significant difference or most significant difference with the HWEM group, respectively (*p* < 0.05 or *p* < 0.01). “A, a” designate *p* < 0.05 or *p* < 0.01 between them.

**Figure 2 antioxidants-10-01689-f002:**
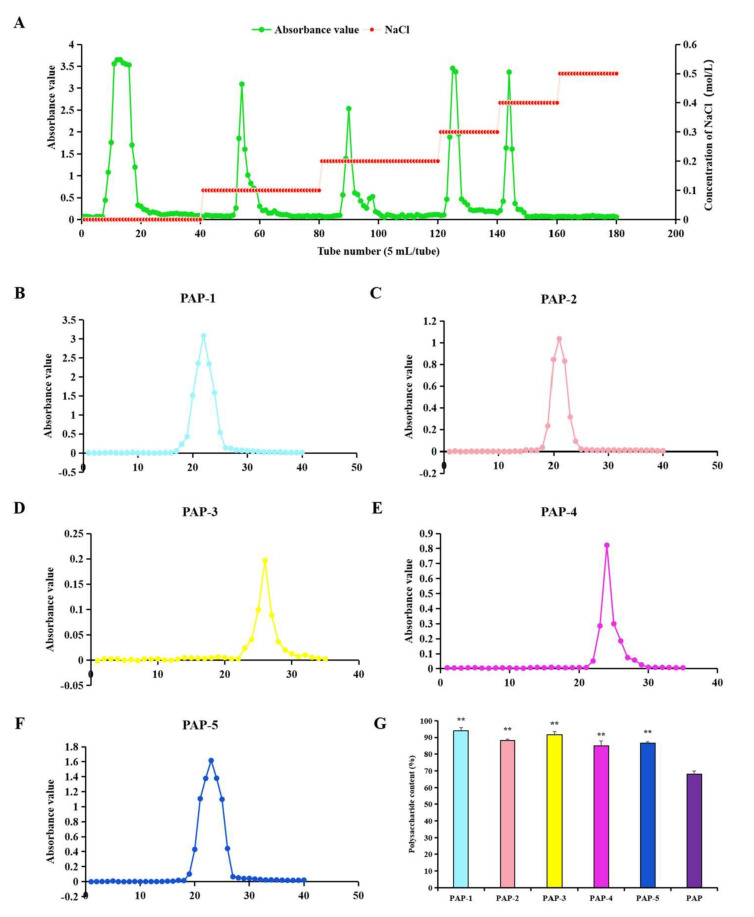
The elution curve of PAP was separated and purified by Cellulose DE-52 chromatography column (**A**) and Sephacryl S-200 high-resolution chromatography column (**B**–**F**) respectively; (**G**) Polysaccharide content of PAPs (means ± SD, *n* = 3). Bars with ** indicates most significant difference with other groups (*p* < 0.01).

**Figure 3 antioxidants-10-01689-f003:**
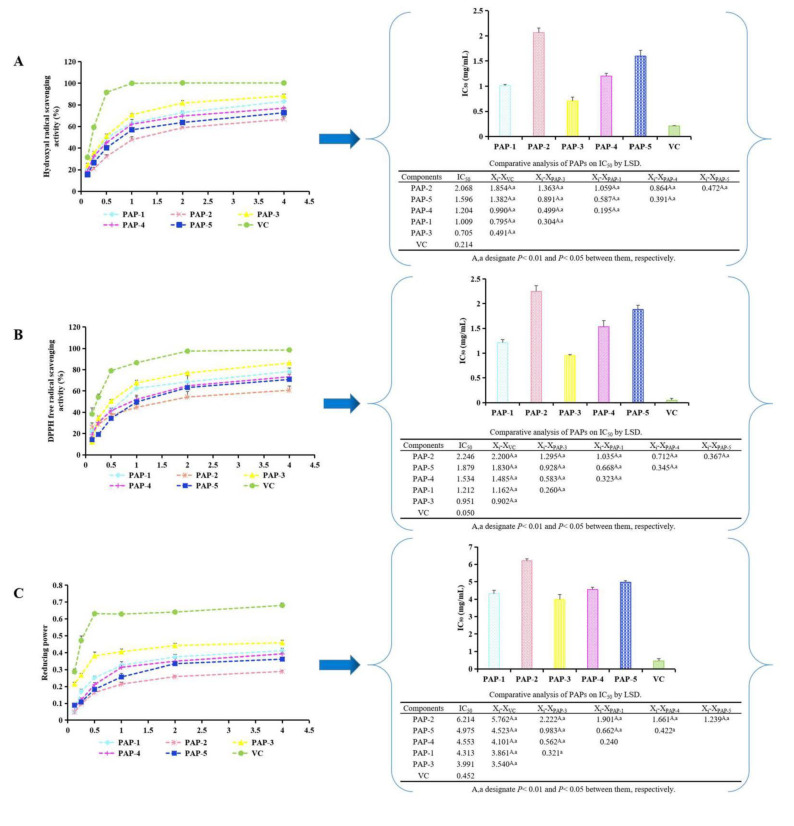
(**A**) Scavenging activity of PAPs on hydroxyl free radicals. (**B**) Scavenging activity of PAPs on DPPH free radicals. (**C**) Reducing power of PAPs. (means ± SD, *n* = 3). “A, a” designate *p* < 0.05 or *p* < 0.01 between them.

**Figure 4 antioxidants-10-01689-f004:**
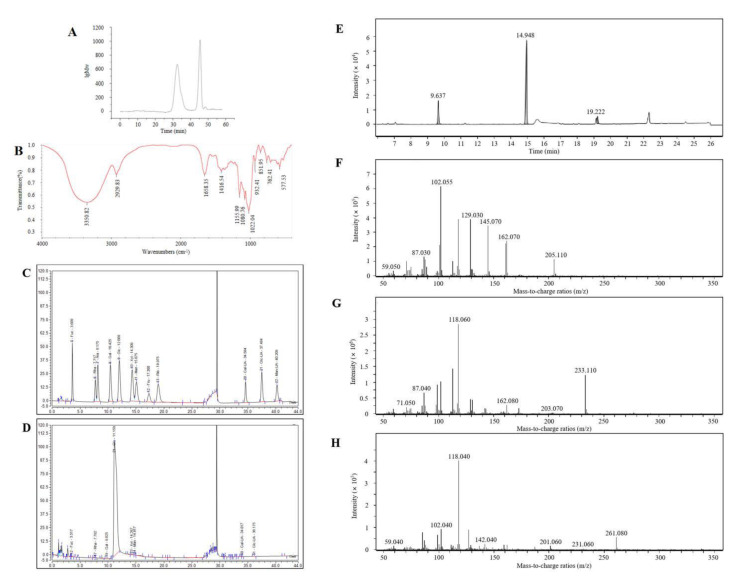
HPGPC chromatogram (**A**) and FT-IR spectrum (**B**) of PAP-1. Chromatogram of standard (**C**) and PAP-1 (**D**).The spectrum of PAP-1 by methylation analysis: (**E**) Spectrum of total ion flow of PAP-1; (**F**) Spectrum of t-Glc(p); (**G**) Spectrum of 4-Glc(p); (**H**) Spectrum of 4,6-Glc(p).

**Figure 5 antioxidants-10-01689-f005:**
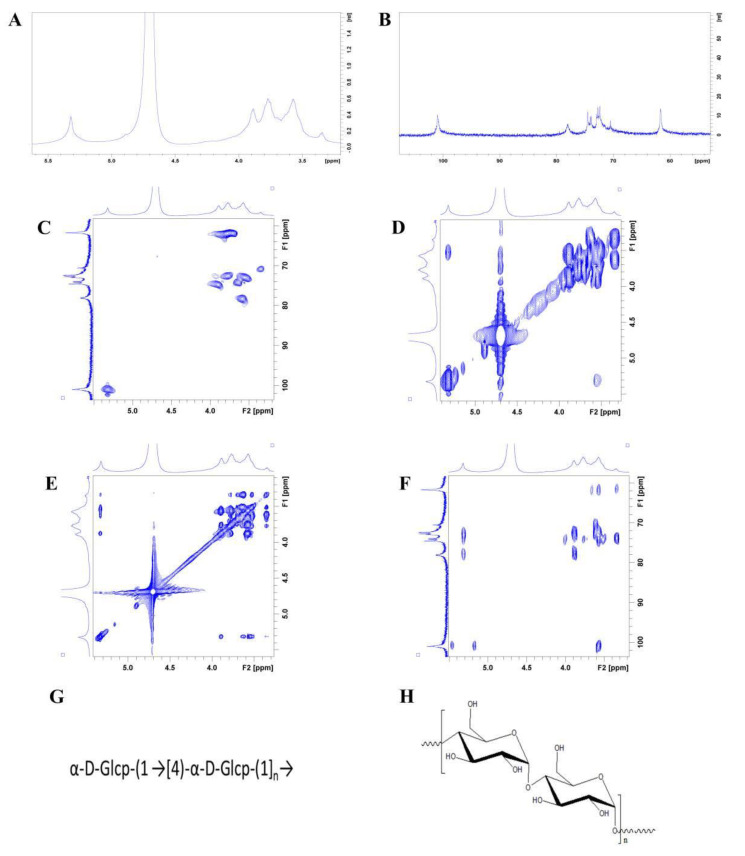
NMR spectrum and structure of PAP-1. (**A**) ^1^H NMR spectrum; (**B**) ^13^C NMR spectrum; (**C**) HH-COSY spectrum; (**D**) HSQC spectrum; (**E**) HMBC spectrum; (**F**) TOCSY spectrum; (**G**) Bonding structures of PAP-1; (**H**) Structural formula of PAP-1.

**Figure 6 antioxidants-10-01689-f006:**
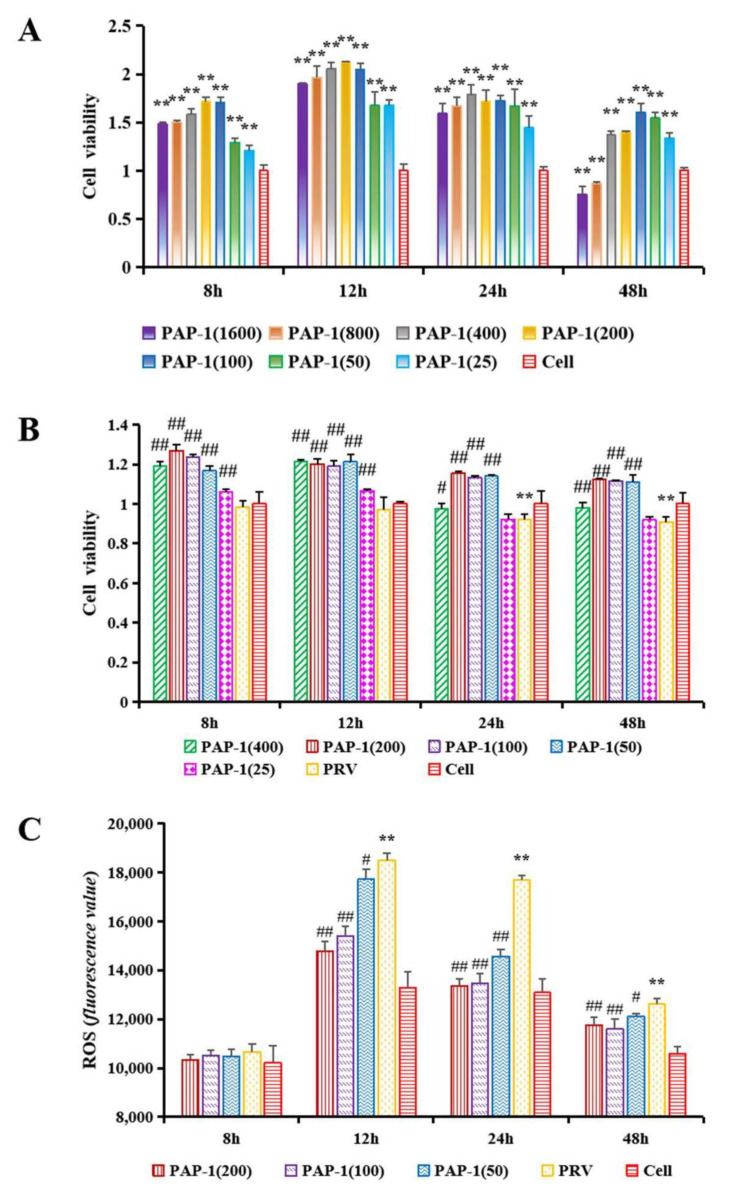
The effect of PAP-1 on the activity of RAW264.7 cells (**A**). The regulation of PAP-1 on the activity (**B**) and ROS level (**C**) in PRV-infected RAW264.7 cells. (means ± SD, *n* = 4). Bars with # indicate a significant difference between the PRV group, respectively (*p* < 0.05). ** and ## show the most significant differences with cell group and PRV group, respectively (*p* < 0.01).

**Figure 7 antioxidants-10-01689-f007:**
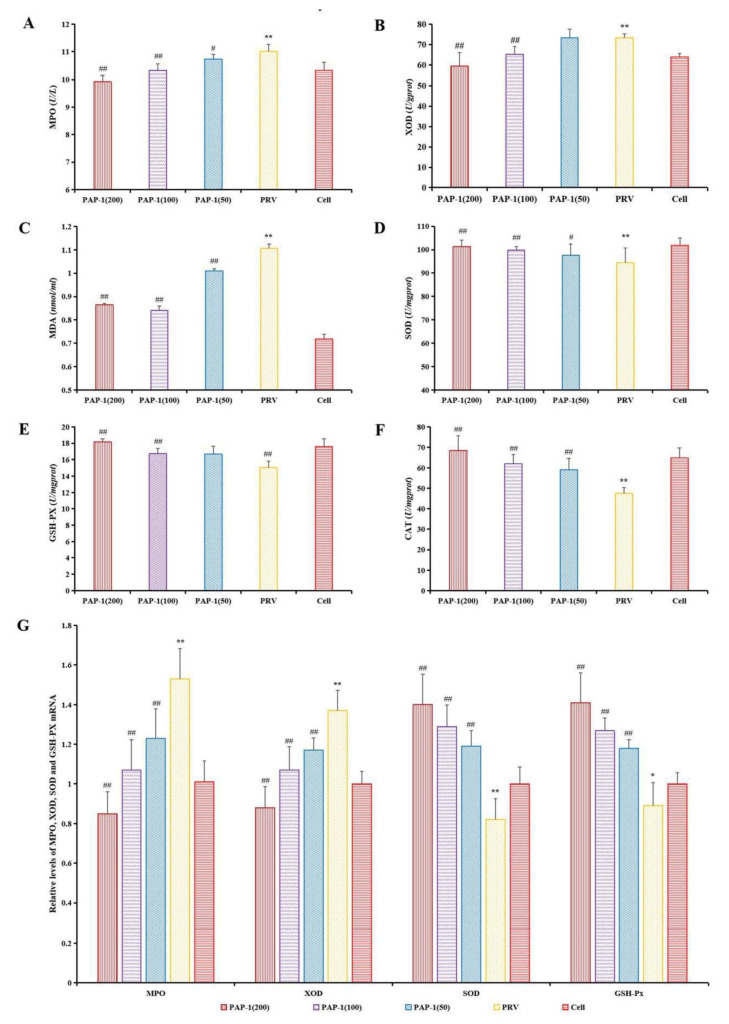
The effect of PAP-1 on the antioxidant activity of PRV-infected RAW264.7 cells. (**A**) The activity of MPO; (**B**) The activity of XOD; (**C**) The level of MDA; (**D**) The activity of SOD; (**E**) The activity of GSH-Px; (**F**) The activity of CAT; (**G**) Relative levels of MPO, XOD, SOD, and GSH-Px mRNA. (means ± SD, *n* = 9). Bars with * and # indicate a significant difference between the cell group and the PRV group (*p* < 0.05). ** and ## show the most significant differences with the cell group and PRV group, respectively (*p* < 0.01).

**Figure 8 antioxidants-10-01689-f008:**
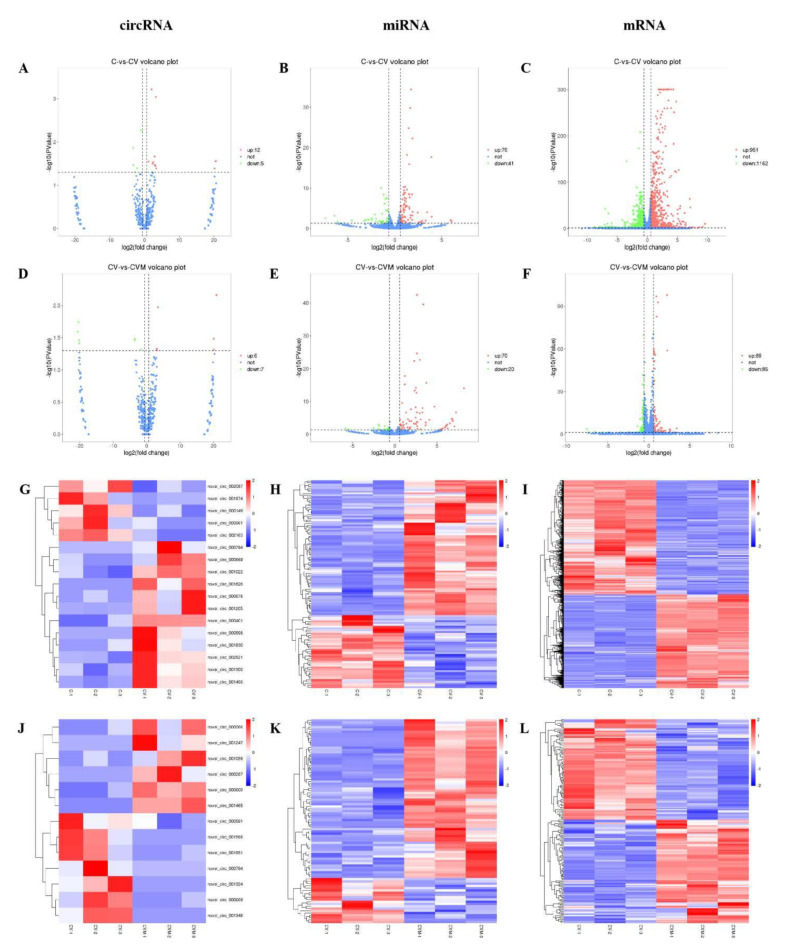
Volcano map and cluster heat map of circRNA, miRNA, and mRNA expression. 17 differentially expressed circRNAs, 117 miRNAs, and 2113 mRNAs between C group and CV group (**A**–**C**) (*p* < 0.05, FC > 1.5). There are 13 differentially expressed circRNAs, 90 miRNAs, and 175 mRNAs between the CV group and the CVM group (**D**–**F**) (*p* < 0.05, FC > 1.5). Heat map generated by hierarchical clustering analysis of differentially expressed circRNA (**G**,**J**), miRNA (**H**,**K**), and mRNA (**I**,**L**).

**Figure 9 antioxidants-10-01689-f009:**
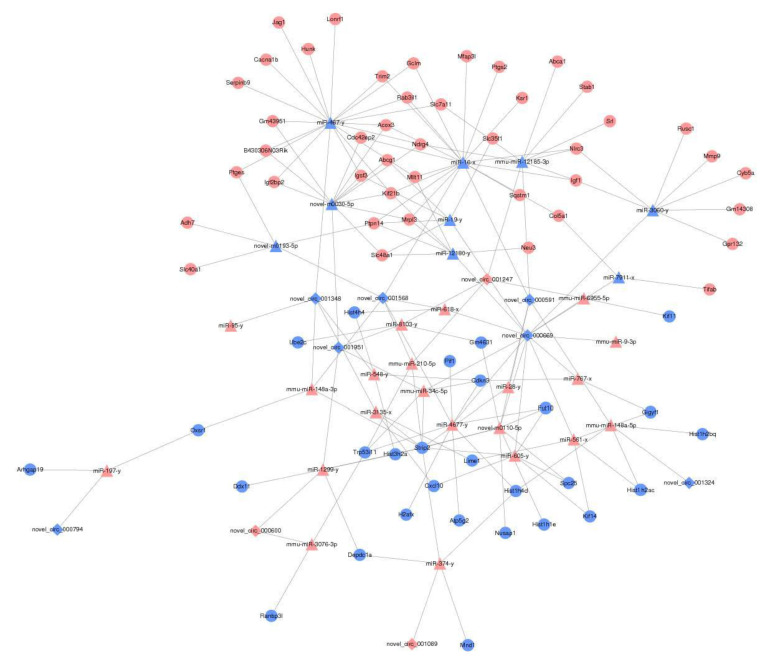
The regulation network of PAP-1 on circRNA-miRNA-mRNA in PRV-infected RAW264.7 cells. The up-regulated mRNA was represented as a red circle, while the down-regulated mRNA was represented as a blue circle. The up-regulated circRNA was represented by a red diamond, while the down-regulated circRNA was represented by a blue diamond. The up-regulated miRNA is represented by a red triangle, while the down-regulated miRNA is represented by a blue triangle. Sequence targeting relationships between miRNA and other RNA analyses are shown in black lines.

**Figure 10 antioxidants-10-01689-f010:**
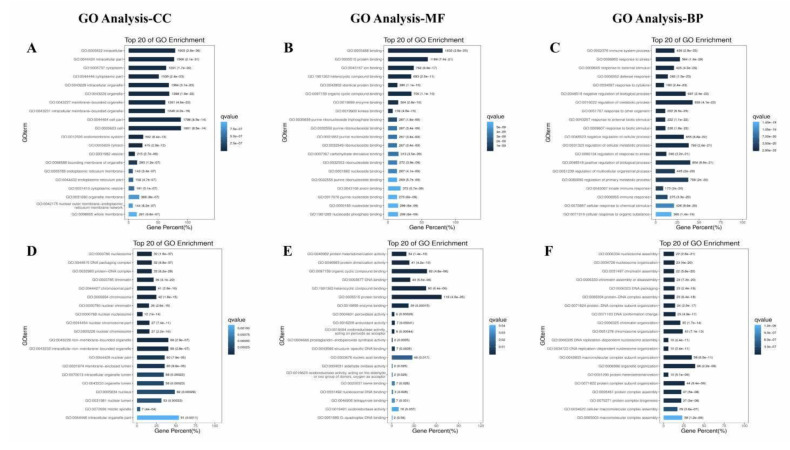
Gene Ontology term analysis of differentially expressed mRNA. C group and CV group: Gene Ontology annotation of mRNA in cellular components (**A**), molecular functions (**B**), and biological process (**C**). CV group and CVM group: Gene Ontology annotation of mRNA in cellular components (**D**), molecular functions (**E**), and biological process (**F**). GO, Gene Ontology; CC, cellular component; MF, molecular function; BP, biological process.

**Figure 11 antioxidants-10-01689-f011:**
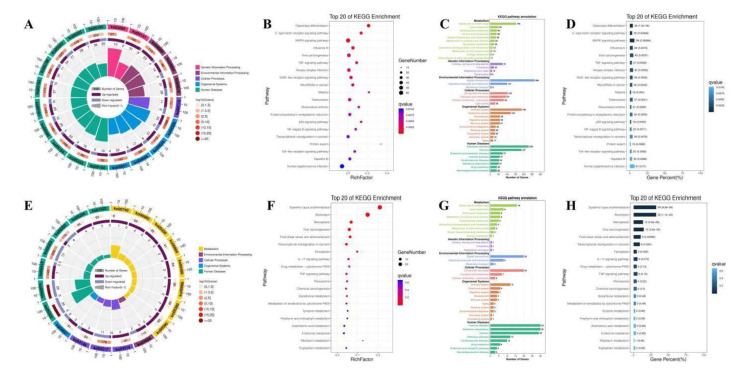
Ko term analysis of differentially expressed mRNA. Enrichment and screening of KEGG pathway in C group and CV group re-sulted in 315 signaling pathways, involving osteogenic class differentiation, c-type lectin re-ceptor signaling pathway, MAPK signaling pathway, Viral carcinogenesis, TNF signal-ing pathway, herpes simplex infection, nod-like receptor signaling pathway, microRNAs in cancer, Malaria et al. (**A**–**D**). 171 signal path-ways were obtained by enrich-ment and screening of the KEGG pathway in the CV Group and the CVM Group, which mainly involved systemic lupus erythematosus, alcoholism, necroposis, viral carcino-genesis, fluid shear stress and atheroslerosis, transcriptional misregulation in cancers, and others signaling pathways (**E**–**H**).

## Data Availability

Data is contained within the article.

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
