# Peer review of "Separation, Purification, Structure Analysis, In Vitro Antioxidant Activity and circRNA-miRNA-mRNA Regulatory Network on PRV-Infected RAW264.7 Cells of a Polysaccharide Derived from Arthrospira platensis"

_antioxidants, 2021, doi:10.3390/antiox10111689_

Round 1

Reviewer 1 Report

In my opinion the paper is worth studying and the manuscript contains enough original and interesting material. It is clearly and concisely written. The experimental procedures are described comprehensively. The results are interesting.

Minor corrections:

Figure 2 is illegible.

The language should be modified carefully.

Text formatting should be carefully checked.

Author Response

Reviewer 1

  1. Figure 2 is illegible.

Response: We adjusted the resolution of Figure 2.

  1. The language should be modified carefully.

Response: We revised the language of the manuscript in detail.

  1. Text formatting should be carefully checked.

Response:We modified the text formatting of the manuscript in detail.

Reviewer 2 Report

English grammar needs to be improved. All document needs to be checked.

In the abstract, the graphic representation of the polymer structure is not necessary; explain it but do not include a figure.

Many abbreviations are not needed. Abbreviations are difficult for readers to follow, avoid their use.

The authors cannot abbreviate polysaccharides from Spirulina platensis as PSP; it is the genus and species scientific name. In fact, the name Spirulina is the commercial one, while the correct is Arthrospira platensis.

It is not clear if this is an intracellular or extracellular polysaccharide; clarify this in the abstract and introduction.

The methodology needs to be written so that the methods are clearly described, and the authors must explain why such measurement is important; what will it demonstrate?

It is not clear what is the added value of the information regarding the circRNA-miRNA-mRNA regulatory 10 network. This needs to be clearly explained.

Specific comments and concerns are found in the attachment as pdf. They need to be  followed.

Author Response

Reviewer 2

  1. English grammar needs to be improved. All document needs to be checked.

Response: We carefully checked and revised the English grammar of all the documents.

  1. In the abstract, the graphic representation of the polymer structure is not necessary; explain it but do not include a figure.

Response: We have removed the picture from the abstract.

  1. Many abbreviations are not needed. Abbreviations are difficult for readers to follow, avoid their use.

Response: We redacted unnecessary abbreviations in the manuscript.

  1. The authors cannot abbreviate polysaccharides from Spirulina platensis as PSP; it is the genus and species scientific name. In fact, the name Spirulina is the commercial one, while the correct is Arthrospira platensis.

Response: We have modified Spirulina platensis to Arthrospira platensis and the abbreviation PSP to PAP.

  1. It is not clear if this is an intracellular or extracellular polysaccharide; clarify this in the abstract and introduction.

Response: PAP extracted in this study is an intracellular polysaccharide, which has been identified in the abstract and preface.

  1. The methodology needs to be written so that the methods are clearly described, and the authors must explain why such measurement is important; what will it demonstrate? “2.7. Statistical analysis” section: why this test and not Tukey? And what about a repeated measures ANOVA?

Response: Since the LSD test has the highest sensitivity, it is essentially a T-test. Usually used to compare one or more pairs of professionally significant sample means. Although Tukey method is also a good multiple comparison method, Tukey test is more conservative than LSD-T test, that is, it is difficult to find significant differences compared with LSD-T test. ANOVA is abbreviation for One-way Analysis of Variance.

  1. It is not clear what is the added value of the information regarding the circRNA-miRNA-mRNA regulatory network. This needs to be clearly explained.

Response: The screening of antioxidation-related genes by circRNA-miRNA-mRNA regulatory network is helpful to further study the effect of PAP-1 on the antioxidant effect of PRV-infected RAW264.7 cells.

  1. Specific comments and concerns are found in the attachment as pdf. They need to be followed.

Response: Specific comments in PDF have also been revised.

  1. Characterization of the extract in terms of carbohydrates, proteins? How do the authors confirm that it's only a polysaccharide? This polysacharide has been previously studied and is called SPIRULAN, search for literature about it.

Response: FT-IR spectrum of PAP-1 showed that PAP-1 had the basic characteristics of polysaccharide. The results of ninhydrin test and UV spectrum showed that PAP-1 contained no protein. As the peak of PAP-1 purified by Sephacryl S-200 high-resolution Gel column is single and symmetrical, it contains only one polysaccharide.

  1. Proteins should have also been quantified.

Response: Because the extracted PAP has been treated with protein removal, no purple reaction was observed by ninhydrin test and no special absorption peak was observed at 280 nm by ultraviolet spectrum scanning. It was preliminarily judged that there was no protein, so protein content was not detected.

  1. “3.4.3. FT-IR” section:(protein, amino groups?)

Response: No protein and amino acid groups were observed in FT-IR spectrum of PAP-1, and no protein and amino acid were found in combination with UV scanning.

  1. Figure 5 is unclear, how is this comparison made? A repeated measures ANOVA? Elaborate explanation.

Response: In Figure 5, the present figure 6, variance analysis was also carried out by SPSS software, followed by LSD-T test for multiple comparison. We compared the other groups to the cell group to show what happened compared to the cell group. In addition, comparing the other groups with the PRV group showed what had changed compared with the PRV group.

Reviewer 3 Report

In the manuscript of M. Cao, et al. “Separation, Purification, Structure Analysis, In Vitro Antioxidant Activity and circRNA-miRNA-mRNA Regulatory Network on PRV-Infected RAW264.7 Cells of a Polysaccharide Derived from Spirulina platensis” isolation, purification, and structural characterization of dextran type polysaccharide named PSP-1 is described. It is composed mainly of alpha-D-(1-4)-glucose units. The antioxidant activity and regulation of PSP-1 on PRV-infected RAW264.7 cells of circRNA-miRNA-mRNA network were investigated by chemical kit, Q-PCR, and ce-RNA seq. It was shown that PSP-1 increased the antioxidant capacity and regulated circRNA-miRNA-19 mRNA network in PRV-infected RAW264.7 cells.

Reviewer’s notes.

  1. All species names should be italicized (from the title, line 5 to the list of references).
  2. Polysaccharides from Spirulina platensis were studied for decades. It is necessary to compare the present structural studies with those done previously, especially with published recently by J. Li, et al. “Isolation, Purification, Characterization, and Immunomodulatory Activity Analysis of α‑Glucans from Spirulina platensis”, DOI: 10.1021/acsomega.1c02175 (not cited here).
  3. Lines 222—223: “The monosaccharide components were detected by an electrochemical detector.” What type of detector was used? (PAD, PD, or else).
  4. Line 236. “PSP-1 (1 mg) was methylated so that all the free hydroxyl groups were methylated.” What procedure (Hakomori, Ciucanu-Kerek) was used? How complete methylation was achieved and approved?
  5. Line 238: “These monosaccharides were restored and acetylated…” Should be: “reduced”, the reduction was done with sodium borodeuteride, as it can be seen from mass spectra.
  6. Lines 242—244. GC-MS conditions are incomplete.

Author Response

Reviewer 3

  1. All species names should be italicized (from the title, line 5 to the list of references).

Response: We change all species names to italics.

  1. Polysaccharides from Spirulina platensis were studied for decades. It is necessary to compare the present structural studies with those done previously, especially with published recently by J. Li, et al. “Isolation, Purification, Characterization, and Immunomodulatory Activity Analysis of α‑Glucans from Spirulina platensis”, DOI: 10.1021/acsomega.1c02175 (not cited here).

Response: We compared PAP-1 in this study with previous studies (Lines 474-487).

  1. Lines 222—223: “The monosaccharide components were detected by an electrochemical detector.” What type of detector was used? (PAD, PD, or else).

Response: The monosaccharide components were detected by the electro-chemical detector (HPLC-PAD) (Line 239).

  1. Line 236. “PSP-1 (1 mg) was methylated so that all the free hydroxyl groups were methylated.” What procedure (Hakomori, Ciucanu-Kerek) was used? How complete methylation was achieved and approved?

Response: The Hakomori method was used in this study. The key of methylation reaction is complete methylation, and infrared spectroscopy is usually used to detect absorption at 3500 cm-1 to judge.

  1. Line 238: “These monosaccharides were restored and acetylated…” Should be: “reduced”, the reduction was done with sodium borodeuteride, as it can be seen from mass spectra.

Response: “These monosaccharides were restored and acetylated…” changed to “These monosaccharides were reduced and acetylated…” (Line 256).

  1. Lines 242—244. GC-MS conditions are incomplete.

Response: We have perfected the conditions of GC-MS. Mass spectrometry conditions were as follows: PAP-1 was detected by electron bombardment ion source (EI) in full SCAN mode, and the mass scanning range (m/z) was 30-600 (Agilent 5977B; Agilent Technologies, USA). Gas chromatography conditions: 140℃ for 2.0 min, 3℃/min to 230℃ for 3 min, and the injection volume was 1 μl (Agilent 7820A; Agilent Technologies, USA) (Lines 260-264).

Round 2

Reviewer 2 Report

Line 109: it is unclear what the authors tell with 5 kg/m2 biomass.

In section 2.5.4 specify the volume of 2M TFA solution added to the 5 mg of polysaccharide.

Line 408. What do the authors refer to with  “a suitable concentration range.” It is not clear.

A clear conclusion(s) or take-home message is missing in section 3.8. What is the importance of the results? Or in what way the results presented contribute?  

The conclusions need to be improved, all the results need to be briefly summarized, and the section should include clear take-home messages. The structure figure is not necessary.

Author Response

Response to the Reviewers’ comments from Tingjun HU et al

Reviewer comments:

  1. Line 109: it is unclear what the authors tell with 5 kg/m2

Response: The production base of Arthrospira platensis can produce 500 tons of high quality Arthrospira platensis powder per 100,000 square meters of breeding area, so we want to express that the biomass of Arthrospira platensis is 5 kg/m2 (Lines 86-87).

  1. In section 2.5.4 specify the volume of 2M TFA solution added to the 5 mg of polysaccharide.

Response: We added the volume of 2M TFA solution in the manuscript: “2 M trifluoroacetic acid (2 mL)” (Line 181).

  1. Line 408. What do the authors refer to with “a suitable concentration range.” It is not clear.

Response: “a suitable concentration range means”: Excessively low concentration of plant polysaccharide may not significantly improve cell activity, while excessive concentration may cause toxicity to cells and inhibit cell proliferation) (Lines 392-394).

  1. A clear conclusion(s) or take-home message is missing in section 3.8. What is the importance of the results? Or in what way the results presented contribute?

Response: We added this paragraph at the end of section 3.8 of the manuscript: “In conclusion, ceRNA-seq showed that PAP-1 increased the expression of antioxidation-related genes in PRV-infected RAW264.7 cells by regulating circRNA-miRNA-mRNA network, and caused changes in antioxidation-related signaling pathways. This study can provide a research basis for circRNA-miRNA-mRNA as a drug target to improve the antioxidant capacity of virus-infected cells.” (Lines 491-494).

  1. The conclusions need to be improved, all the results need to be briefly summarized, and the section should include clear take-home messages. The structure figure is not necessary.

Response: The conclusion section has been improved and all the results sections have been briefly summarized with clear key information. In addition, the structure diagram of PAP-1 in the conclusion has been deleted.

Round 3

Reviewer 2 Report

The current version is suitable for publication